# Pain Neuroscience Education and Neuroimaging—A Narrative Review

**DOI:** 10.3390/brainsci14090947

**Published:** 2024-09-22

**Authors:** Daniele Corbo

**Affiliations:** Department of Medical and Surgical Specialties, Radiological Sciences and Public Health, University of Brescia, 25123 Brescia, Italy; daniele.corbo@unibs.it

**Keywords:** PNE, neuroimaging, neurophysiology, neurosciences, pain

## Abstract

Background: Musculoskeletal pain is a leading cause of medical visits, posing significant challenges both socially and economically, encouraging the scientific community to continue researching and exploring the most effective methods to address the problem. An alternative way to deal with chronic pain is pain neuroscience education (PNE), a lesson plan that addresses the neurobiology, neurophysiology, and nervous system processing of pain. This method takes the place of the conventional one, which connected pain to tissue damage or nociception. Results: As a result, patients are taught that pain is often not a reliable measure of the health of the tissues but rather the outcome of the nervous system interpreting the injury in conjunction with additional psychosocial variables. In addition to finding research that examine, using neuroimaging, whether the administration of PNE has detectable effects at the level of the central nervous system, this narrative review seeks to clarify what PNE is, how it is administered, and if it is an effective treatment for musculoskeletal pain. Conclusions: Based on the findings, it appears that PNE is more therapeutically beneficial when combined with therapeutic exercise, when done one-on-one, and during lengthy, frequent sessions. Lastly, even though PNE has no effect on the morphological properties of the gray matter, it appears to cause decreased activation of the regions linked to pain.

## 1. Introduction

Musculoskeletal pain is among the main reasons patients visit physiotherapists and doctors, and represents a problem from both a social and economic point of view [1,2]. This is especially true for persistent musculoskeletal pain, which is estimated to cost between $560 and $635 billion in medical costs and lost productivity [3]. It is expected that these numbers may undergo a sharp increase in the future [4], prompting the scientific community to reflect and continue to think and test, through research, the most efficient way to combat them. Patient education is part of the guidelines for the treatment of musculoskeletal disorders. People need education about pain [5], to get answers as to why they are no longer able to move and do what they used to without experiencing any symptoms. Education is therapy and knowledge is therapy. Pain neuroscience education (PNE) is a lesson plan that covers the neurobiology, neurophysiology, and nervous system processing of pain [6]. PNE aims to explain how the nervous system interprets information from the tissues through peripheral nerve sensitization, central sensitization, synaptic activity, and brain processing, and that neural activation, as either upregulation or downregulation, has the ability to modulate the pain experience [7]. Often, teaching people suffering from pain about the anatomy and science of pain can increase anxiety, fear, and false beliefs and consequently increase the sensation of pain [8]. PNE aims to reconceptualize an individual’s understanding of their pain as less threatening to facilitate rehabilitation [9]. This approach replaces the traditional model of linking tissue injury or nociception and pain [10]. Furthermore, a 100 to 200 min dose of PNE is estimated to be optimal for kinesiophobia and anxiety symptoms, while the optimal dose for catastrophizing is 400 min. Patients are therefore taught that pain is not always an accurate indicator of the state of the tissues and that it is instead the result of the nervous system processing the damage in combination with other psychosocial factors [6,11,12]. Successful PNE is linked to the complexity of each person’s individual processing of pain [13]. Patients may be more likely to move, exercise, and push through some discomfort if they reframe their pain as the nervous system’s perception of the threat of the injury rather than an actual indicator of the severity of the cause of the pain [10]. PNE seems to be a much-debated topic in the literature in recent years. In fact, there are numerous works that analyze the topic, even if today the most used type of education at a clinical level regarding pain remains the biomedical/biomechanical model, which mainly emphasizes tissues and tissue lesions as a cause of physical problems [14,15,16]. The first aim of this narrative review was to understand what PNE is, how it is delivered, and whether it is an effective treatment for musculoskeletal pain. The second was to find studies that analyze, through neuroimaging, whether the administration of PNE has measurable effects at the central nervous system level.

## 2. Materials and Methods

### 2.1. Study Experimental Design

This narrative review was conducted following a re-adaptation of the PRISMA flow [17]. The research was carried out in the period between September 2023 and June 2024.

### 2.2. Inclusion Criteria

The purpose of this narrative review is twofold. First of all, include all the summary results, i.e., the secondary studies, present in the literature on the effectiveness of pain neuroscience education (PNE) in the treatment of musculoskeletal pain. To be included, research studies had to be written in English, present in full text, and provide sufficient methodological details regarding the type of PNE treatment either alone or in combination with other treatments. No age or publication year restrictions were applied to the included samples. Secondly, all research studies that met the following criterion were included: neuroimaging studies investigating the effects of PNE in patients with musculoskeletal problems.

### 2.3. Study Selection

In order to evaluate the relevant literature in the field, works published up to June 2024, chosen from a search conducted in open databases (PubMed, Google Scholar, WoS, Scopus), were included in this qualitative critical review. The initial search and all databases used a combination of the following terms: [(“Pain neuroscience education” OR “Therapeutic neuroscience education” OR “Neuroscience education” OR “Pain neurophysiology education” OR “Pain education” OR “Explain pain” OR “ PNE” OR “TNE” OR “PE” NOT “Participatory ergonomics” NOT “Pulmonary embolism” NOT “*Percutaneous* needle electrolysis”) AND (“Musculoskeletal pain” OR “Fibromyalgia” OR “Chronic pain” OR “Osteoarthritis” OR “Low back pain” OR “Neck pain”)] without specifying a certain period of publication dates. Originally 1065 studies were identified through the database search. Once duplicates were removed and the titles and abstracts of all remaining unique articles were analyzed, 166 full-text articles were analyzed to verify their eligibility for inclusion in the present study. Of these 166, 139 of these articles were excluded, so 27 studies were finally selected for the last check, in which 5 of these were excluded. The total number of articles that met the inclusion criteria was 22. As regards the second research question, the search was conducted in open databases (PubMed, Google Scholar, WoS, Scopus). All databases used a combination of the following terms: [(“Pain neuroscience education” OR “Therapeutic neuroscience education” OR “Neuroscience education” OR “Pain neurophysiology education” OR “Pain education” OR “Explain pain” OR “Pain physiology education”) AND (“Neuroimaging” OR “Imaging” OR “MRI” OR “Magnetic resonance imaging” OR “fMRI” OR “Functional magnetic resonance imaging”)]. We found 65 results from our initial search, among which, after reading titles and abstracts, we discarded 61. The total articles that met the inclusion criteria were 4. Inclusion, exclusion and final inclusion are summarized in the flowchart (Figure 1).

## 3. Results

### 3.1. Clinical Results

All clinical results were summarized in Table 1.

A systematic review by Louw and colleagues [6] analyzed the effects of PNE on pain, disability, anxiety, and stress in patients with chronic musculoskeletal pain and is the first systematic review to investigate these measures in this type of population [18]. The review is based on eight studies (including six RCTs, one pseudo-RCT, and one comparative study; number of participants = 401). The study concluded that the results indicate compelling evidence for the use of PNE in reducing pain rates, increasing physical performance, decreasing perceived disability, and decreasing catastrophizing in patients with chronic musculoskeletal pain. The previous study was updated by a new systematic review by Louw and colleagues [7], which investigated the effectiveness of PNE on musculoskeletal pain. The review found that PNE is effective in reducing pain, improving patient knowledge of pain, and improving function and reducing disability in individuals (participant number = 734) with chronic musculoskeletal pain. They included 13 RCTs and assessed the quality of the studies using the PEDro scale. It is also important to recognize that no PNE study has shown worse outcomes than control groups, implying a significant benefit-risk balance in favor of PNE. Furthermore, three studies reported results one year after the start of treatment and all these studies showed a significant reduction in the request for support from health professionals and therefore, consequently a reduction in healthcare costs [8,19]. A systematic review and meta-analysis investigated the clinical effectiveness of PNE for people with chronic musculoskeletal pain [20]. This mixed review, which included 12 RCTs (number of participants = 755) and four qualitative studies (number of participants = 50), demonstrated that PNE can reduce pain, disability, catastrophizing, and fear of movement. In the short term, pain reduction and disability appear to be of little clinical relevance. In contrast, the results regarding medium-term pain catastrophizing and short-term reduction of fear of movement led to clinical significance. Greater and clinically relevant effects on pain (short- and medium-term), disability (medium-term), and catastrophizing were observed when PNE was associated with another intervention compared to PNE administered alone. Larger effects for disability and catastrophizing (medium-term) were observed when longer sessions of PNE were administered. Data from the four qualitative studies identified several key components for improving the patient experience with PNE, such as allowing the patient to tell their story. These components may improve pain reconceptualization, which appears to be an important process in facilitating patients’ abilities to cope with their condition. Watson and colleagues [9] conducted a further systematic review and meta-analysis to investigate whether there were interindividual differences in PNE treatment in adults with musculoskeletal pain, so that the intervention could be tailored to individuals to optimize its effectiveness. The review included five RCTs (number of participants = 428). The findings currently provide little evidence of true variation in people’s response to PNE for pain, disability, and psychosocial outcomes in adults with musculoskeletal pain. Bülow and colleagues [21] investigated the effectiveness of PNE on pain, disability, and psychological distress in musculoskeletal pain. The review included 18 RCTs (number of participants = 1585). The conclusions following this study were that PNE has statistically significant, low to moderate effects on pain intensity, disability, and short-term psychological distress. Furthermore, there was a significant effect on pain intensity in long-term follow-ups. However, the effects of PNE in chronic musculoskeletal problems were moderate and statistically significant on pain intensity and psychological distress at both time points, both post-intervention and follow-up. A systematic review and meta-analysis examined the short-term (<12 weeks) impact of combining PNE with exercise for chronic musculoskeletal pain [22]. The review included five high-quality RCTs (number of participants = 460) comparing the combination of PNE with exercise alone. The study found that combining PNE with exercise resulted in greater short-term improvements in pain, disability, fear of movement, and pain catastrophizing than exercise alone in individuals with chronic musculoskeletal pain. A recent systematic review and meta-analysis found that PNE had small to moderate effects on pain, disability, and psychological distress in musculoskeletal pain [23]. The review included 15 RCTs (number of participants = 1085). In particular, it found the effects of one-to-one sessions to be better than remote sessions (telephone and computer) or content-only reading in improving pain, disability, and psychosocial factors in patients with musculoskeletal pain. Furthermore, they noted that in patients with fibromyalgia, PNE has shown promising results when included in a multidisciplinary program compared to usual care, but not compared to other educational or self-management techniques. In patients with chronic spinal pain and chronic fatigue syndrome, PNE appears reliable for improving short-term clinical outcomes when proposed individually compared to other educational or self-management approaches. In patients with chronic low back pain, it appears to be effective in combination with other treatments, such as manual therapy and especially therapeutic exercise. De Oliveira Lima and colleagues [24] investigated the effectiveness of PNE carried out on the web without clinical support compared to minimal intervention (no intervention or booklets) for the intensity of pain in the short and medium term. The studies that met the inclusion criteria were six RCTs (number of participants = 1664). The duration of the web program ranged from 3 to 8 weeks in isolation and from 4 to 6 months in combination with usual care. Web-based pain education for adults with musculoskeletal pain has been shown to be slightly better than minimal intervention for pain intensity, disability, kinesiophobia, and short-term global impression of change, as well as for pain intensity and global impression of change in the medium term. Furthermore, the intervention was better than usual care alone for medium-term global impression of change, but did not provide additional benefits for other medium- or long-term primary and secondary outcomes. A meta-analysis analyzed the overall effect of pain neuroscience education on chronic musculoskeletal pain and assessed whether this effect differs based on dosage and other components of the treatment format [25]. Dosage included the number of PNE sessions and the amount of time per session. Structural components included PNE provided alone as a treatment or combined with other pain management modalities, as well as the inclusion of group treatment sessions. Eighteen studies were included in this meta-analysis. The current meta-analysis found that PNE is a significant treatment modality for Improving the outcome of pain intensity, disability, pain catastrophizing, and fear of movement. When exploring specific moderators such as dosage, format, and structure of PNE treatment, there appear to be no significant differences in moderating pain outcome, with the exception of group interventions for fear of movement, which suggest an improvement in this outcome compared to other modes. Salazar-Méndez and colleagues [26] investigated the optimal duration of pain neuroscience education to improve psychosocial variables in chronic musculoskeletal pain. A linear relationship was observed between longer duration of PNE (total minutes) and reduced symptoms of anxiety, catastrophizing, and fear of movement, but was statistically significant only for the catastrophizing variable. Furthermore, a 100 to 200 min dose of PNE is estimated to exceed the minimal clinical change in treatment outcome described in the literature for kinesiophobia and anxiety symptoms, respectively, while the estimated optimal dose for catastrophizing was 400 min. There are numerous systematic reviews of the literature that evaluate how PNE acts in specific musculoskeletal conditions such as chronic low back pain, chronic pain, chronic non-specific spinal pain, osteoarthritis, fibromyalgia, persistent tendinopathies, and chronic neck pain. Clarke and colleagues [27] analyzed whether PNE could be used in the treatment of individuals with chronic low back pain. It is the first work to specifically investigate the evidence of PNE in this type of patient. Two RCTs are included in the review. Studies have shown a significantly better effect on pain for PNE compared to control for up to 12 months. Furthermore, the study suggests that PNE is a promising intervention for physical function, psychological function, and social function. Tegner and colleagues [28] evaluated the effect of PNE for patients with chronic low back pain. The review included seven studies. The main finding of this review was moderate-quality evidence that PNE has an effect on pain relief for patients with chronic low back pain. Furthermore, it found low-quality evidence showing that PNE has an effect on disability immediately after surgery and on pain and disability 3 months after surgery. Wood and Hendrick [29] included eight RCTs (number of participants = 615) in their review. They showed that moderate-quality evidence is provided for the use of PNE to usual physiotherapy interventions in improving disability and performance scores in chronic short-term low back pain. In the systematic review and meta-analysis by Shin and colleagues [30], in which nine RCTs were included (number of participants = 1019), the carryover effects of PNE on pain intensity and cognition (catastrophization and fear) were analyzed in individuals with chronic non-specific low back pain. The results were that the intervention had both a short- and long-term carryover effect on the outcomes examined. A recent systematic review and meta-analysis by Ma and colleagues [31] assessed how pain neuroscience education could impact chronic low back pain in the short term. Seventeen randomized studies (number of participants = 1078) are included in the review, of which 16 are included in the quantitative analysis. According to the meta-analysis, adding PNE to treatment programs could significantly improve short-term pain, disability, and psychological factors (e.g., kinesiophobia and pain catastrophizing) compared to control groups. Furthermore, subgroup analysis based on different PNE strategies found that PNE plus exercise and PNE plus physiotherapy have a greater effect on reducing pain intensity, disability, and pain catastrophizing than PNE, physiotherapy, and exercise alone. A systematic review on chronic pain investigated the effects of education to facilitate knowledge of chronic pain in adults. Nine studies were included, of which PNE was also part of the intervention treatment. The study found no evidence of an effect on pain intensity. However, regarding disability, a significant improvement was found immediately after a PNE training course. A similar effect was not found for the other types of education examined in the studies [32]. Another systematic review and meta-analysis on chronic pain [37] included 14 studies (number of participants = 1024), and investigated the impact of combining pain education strategies (PNE and cognitive behavioral therapy) with physical therapy interventions. The result obtained was a significant improvement in the intensity of pain and disability (both short- and long-term) in the intervention group, i.e., the education and physical therapy group, compared to the control groups which included the following: waiting list, medical management, and traditional physical therapy-only strategies. A systematic review included five RCTs and eight qualitative studies to investigate whether PNE was effective in reducing chronic non-specific spinal pain [33]. The results of this study showed that PNE combined with exercise were generally superior to other forms of intervention such as exercise therapy or multimodal physiotherapy for improving pain, disability, fear of movement, and catastrophizing in the short and medium term. These results were more evident in the medium term (3 to 12 months) than in the short term (<3 months). In the systematic review by Ordoñez-Mora and colleagues [34], in which four RCT articles were included, the effects of PNE on patients with osteoarthritis were investigated. The results suggest that there is an improvement in the PNE group compared to the control, but this cannot necessarily be attributed to the PNE, as small effects were found for variables such as catastrophizing and fear of movement. Saracoglu and colleagues [35] included four RCTs, in which PNE was taken as adjuvant therapy in patients with fibromyalgia syndrome. In a multimodal approach, it was observed that the intervention that included PNE had a greater effect on functional status, pain intensity, catastrophizing, and depression in patients with fibromyalgia. Despite the limited number of studies, the high quality of these indicates that the results could be considered reliable. A recent systematic review investigated the effectiveness of PNE in persistent tendinopathy [38]. Four articles are included in the review, including two RCTs, a randomized feasibility study and a single cohort feasibility study (number of participants = 164). The application of pain neuroscience education in patients with tendinopathies, along with other interventions, appeared to improve several outcomes, including pain, physical performance, function (self-reported), pain catastrophizing, fear of movement, and the perception of illness. However, a study comparing PNE with traditional pathoanatomical education found no clinically significant effects. A recent systematic review and meta-analysis assessed how PNE might impact chronic neck pain [36]. In order to construct this secondary study, seven RCTs were included (number of participants = 478). The results obtained from this meta-analysis demonstrated that PNE significantly reduced pain and decreased kinesiophobia in patients with chronic neck pain compared to control treatment. The benefit was significant in the adult group compared to the adolescent group. The therapeutic effects were not altered regardless of the application of PNE alone or in combination with other treatments.

### 3.2. Neuroimaging Results

Finally, studies were carried out to evaluate whether PNE had an effect on changing the morphology of the brain or its functionality. Figure 2 shows all regions involved by PNE from all studies.

Two case reports reported changes in brain activation in a functional magnetic resonance imaging (fMRI) scan, before and after the application of a PNE program. An RCT performed on 120 samples showed no significant morphological differences in gray matter. In the case study by Moseley [13], it was analyzed in a patient with chronic and disabling low back pain whether PNE could change the brain activation of some areas at the cortical level during the execution of a motor task. For the study, “abdominal suction” was used as an exercise (usually used to activate the transversus abdominis in the supine position), a non-painful task for the patient. Four scans were performed: the first was performed at rest; the second was performed, after training of the exercise by the physiotherapist, while the patient performed the task, with “on” moments of 6 s and “off” moments of 10 s for 10 series; the third was performed 1 week after this, during which the patient was invited to perform the exercise during his daily life; and the last MRI was administered immediately after PNE was performed “one by one” with the patient directly following the third MRI. The result was that the last fMRI, after treatment with PNE, marked a reduction in cortical activation throughout the primary somatosensory cortex. Furthermore, it showed no activation of the cingulate, frontal, or insular areas, components of the so-called “pain matrix”.

In the case study by Louw and colleagues [39], a patient with chronic recurrent low back pain, with an acute episode ongoing for 3 months, was examined. They asked the patient to perform painful motor tasks and have functional MRIs. Four scans were taken for the study: the first was acquired before the session, while the patient was relaxing and watching a cartoon; the second scan was performed at rest; the third scan was performed during a painful physical activity, the anterior pelvic tilt, performed for 30 s of contraction with 30 s of rest alternating for five series; and finally, a 30 min PNE session was performed, after which she repeated the exercise done during the third scan while the fourth fMRI was performed. The post-PNE scan revealed three marked differences compared to the pre-education scan, namely deactivation of the periaqueductal gray and cerebellum, allowing for greater cortical activation in the motor cortex, which may be associated with restored and normal motor activation. A randomized clinical trial was performed, examining 120 individuals with chronic spinal pain to evaluate whether there could be changes in the cortical gray matter thickness of 10 a priori-selected cortical regions and the gray matter volumes of five a priori-selected regions of the subcortical segmentation, following PNE treatment and cognitive training aimed at motor control compared to standard physiotherapy treatment [40]. These selected brain areas in the study showed alterations after conservative treatments in populations with chronic musculoskeletal pain [41]. Both groups had 12 weeks of intervention. No substantial increases (or decreases) were found in any of the brain areas assessed in either intervention [40]. Murillo et al. [42] used both cross-sectional and longitudinal voxel-based morphometry to identify potential gray matter (GM) alterations in people with chronic pain (n = 63) compared to age- and sex-matched pain-free controls (n = 32). They also wanted to know if these GM alterations could be reversed after PNE + exercise (compared to conventional physiotherapy). According to a cross-sectional whole-brain study, those who experience chronic pain have smaller GM in their left inferior temporal gyrus and right and left dorsolateral prefrontal cortex, which is correlated with increased pain vigilance. Additionally, the longitudinal whole-brain analysis showed that after the PNE + exercise treatment, those with chronic pain had decreased GM volumes in the left and right central operculum as well as the supramarginal following treatment. GM in the right dorsolateral prefrontal cortex also rose. Some of these changes were not seen in pain-free controls over time, and they were not exclusive to any one treatment approach (Table 2).

## 4. Discussion

The present review aimed to conduct an in-depth investigation into the literature sources to examine the evidence regarding the application of PNE as a therapy for the treatment of musculoskeletal pain. The analysis of the data and research examined during this review has overall convincingly confirmed the effectiveness and safety of PNE in the context of relief of pain intensity, improvement of disability, optimization of physical function, the mitigation of pain catastrophizing, the reduction of anxiety and stress, as well as the alleviation of psychological distress [6,7,20,21,26]. No adverse events were reported, suggesting that PNE has no contraindications. It is important to point out that no study showed worse results of the experimental treatment compared to the control group, which implies a significant risk/benefit ratio in favor of PNE. Furthermore, it can be analyzed how PNE in addition to other interventions has more effective results in the management of musculoskeletal pain. Although the benefits of physical activity have been demonstrated in various pathologies [43,44,45,46], a review [22] found that PNE, in addition to physical exercise, is more effective than exercise alone, demonstrating significant differences in pain intensity, disability, kinesiophobia, and in the catastrophization of pain. There are also studies in the literature that evaluated whether PNE without clinical support was effective in reducing the intensity of pain, disability, and kinesiophobia. According to the review by De Oliveira and colleagues [24], PNE carried out on the web without clinical support compared to minimal intervention (no intervention or only leaflet) was slightly better than short-term control on pain, disability, and kinesiophobia and on intensity of pain in the medium term [24]. Lepri and colleagues [23], however, demonstrated how the one-to-one modality with reinforcements (brochures and comprehension exercises) seems to be more effective than remote sessions (telephone or computer) or reading content alone. Regarding the posology of PNE, reviews have investigated whether modulating the dosage, format, or structure has led to differences in outcomes. According to Romm and colleagues, there appear to be no significant differences in moderating pain outcome, with the exception of group interventions for fear of movement, which suggest an improvement in this outcome compared to other modalities [25]. According to Salazar-Méndez and colleagues [26], however, a linear relationship was observed between longer duration of PNE (total minutes) and reduction in symptoms of anxiety, catastrophizing, and fear of movement, but it was statistically significant only for the catastrophizing variable. It was also investigated whether PNE might be effective for certain types of people, which would imply some individual differences in response to PNE. There does not currently appear to be any relevant evidence regarding variation in people’s individual response to PNE [20]. Some reviews find evidence of effectiveness of PNE in the treatment of many chronic pain pathologies with regards to pain intensity, fear of movement, psychosocial function, catastrophizing, and kinesiophobia, both in the short and long term [27,28,29,30], chronic pain for disability [32], and effectiveness in pain intensity and disability in both the short and long term [37]. Finally, this review assessed whether there were any studies in the literature that examined whether PNE had an effect on changing brain structure and its functionality. First, it was found to have an impact on brain activity. This means that PNE can affect how the brain processes and interprets pain signals. This effect is relevant because it suggests that it may help people develop a better understanding of pain and learn strategies to manage it more effectively. A major part of the included studies demonstrated changes in the motor area after PNE. This is important because it indicates that the reduction of pain sensation allows the brain to better process motor signals, so the effect of PNE can be seen in the ability to perform movements. Surprisingly, however, no reductions were reported in activation of the insula, typically involved in the imagination and processing of pain. However, it should be noted that despite the effect of brain activation, these studies did not detect morphological differences in the gray matter of the brain. In other words, education about the neuroscience of pain appears to impact brain function, but does not appear to result in physical changes in the structure of the brain in terms of the amount of gray matter. This result is expected, as physiotherapy interventions generally have an impact in terms of brain function, rather than structure. These findings are relevant as they suggest that pain education may be an important element in pain management, especially for people with chronic or persistent pain [13,39,40]. Longer studies with more patients would be interesting to verify whether a longer duration of PNE leads to greater changes at the brain level, also in terms of structure. The greatest limitation of current PNE research concerns the heterogeneity in the various randomized controlled trials. Indeed, there is great diversity in the design, participant inclusion criteria, outcome measures, delivery and delivery methods of PNE, competence of physiotherapist educators, and control interventions in the various systematic reviews. There are wide confidence intervals due to variations in the frequency and duration of interventions from the included studies, as well as variability in the frequency and duration of both exercises and PNE sessions. Furthermore, the large heterogeneity may have influenced the analysis to overestimate the effect sizes. By standardizing these factors, subsequent reviews and meta-analyses could shed further light on the effectiveness of PNE. Its strengths are that this type of educational intervention can be useful in a clinical context as it is simple to combine with other interventions, does not require equipment, and has no side effects for the patient. Interestingly, when administered as a supplement to other interventions this can produce a significant change in pain and disability. Furthermore, it may be relevant to evaluate the cost-effectiveness of PNE offered to patients with chronic musculoskeletal problems, with a view to reducing healthcare costs used for other types of therapies in order to reduce pain.

## 5. Conclusions

Overall, it appears that pain neuroscience education, as a modality within a pain management program, has a significant impact on all pain outcome measures and is, therefore, a valuable intervention for physicians and physical therapists in the treatment of patients with musculoskeletal pain. In particular, it seems to be more clinically effective if added to therapeutic exercise, in one-to-one mode, and with long, high-frequency sessions. However, the results of this review should be interpreted with caution as heterogeneity between studies was high. Finally, it seems that PNE induces less activation of the areas responsible for pain, even if it does not influence the morphological characteristics of the gray matter. Further studies, especially about activation with fMRI, need to be performed to verify previous findings in larger populations, with more rigorous inclusion and application criteria. It is also necessary to use other methods such as PET and electrophysiological study with EEG to verify the brain response to PNE.

## Figures and Tables

**Figure 1 brainsci-14-00947-f001:**
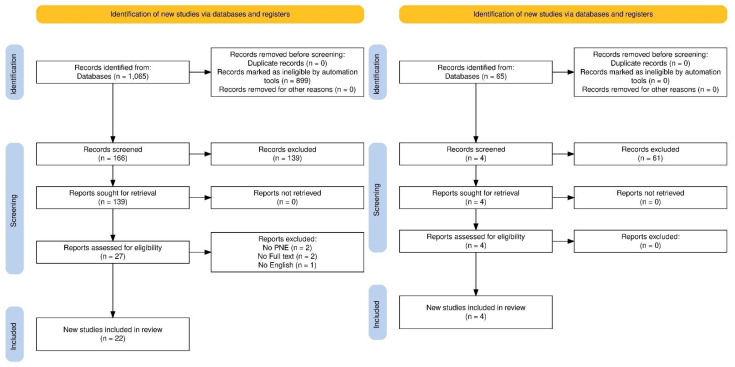
Flowchart of the narrative review process.

**Figure 2 brainsci-14-00947-f002:**
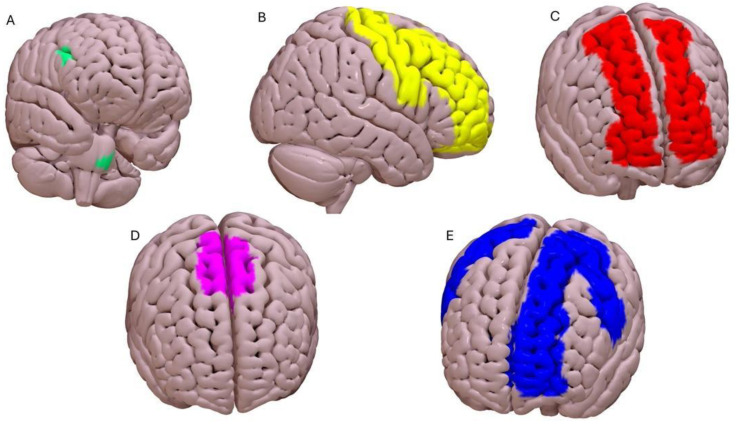
Evidence from neuroimaging studies: (**A**) Basal Ganglia; (**B**) Right Dorsal Prefrontal Cortex; (**C**) Frontal Cortex; (**D**) Motor Cortex; (**E**) Anterior Cingulate Cortex.

**Table 1 brainsci-14-00947-t001:** Summary of all clinical studies.

Author	Participant	Age	Gender (F)	Diagnosis	Diagnostic Criteria	Outcomes
[18]	401	38.2	252	Low back pain, chronic fatigue syndrome, widespread pain, chronic whiplash-associated disorders	WAD I-II according to Quebec Task Force	Positive effect on pain, disability, catastrophization, and physical performance.
[7]	398	41.7	278	Low back pain, chronic fatigue syndrome, fibromyalgia, lumbar radiculopathy awaiting lumbar surgery, chronic neck pain	1990 American College of Rheumatology (ACR) criteria	Reducing pain and improving patient knowledge of pain, improving function and lowering disability, reducing psychosocial factors, enhancing movement, and minimizing health care utilization.
[8]	17	47.2	3	Lumbar surgery	NA	Minimizing the use of provocative terminology could decrease fear, anxiety, and patient pain experiences following lumbar surgery.
[19]	94	42	48	Non-specific chronic low back pain	Oswestry Disability Index	Reducing pain, disability, fear beliefs, mood, and sick leave at long-term follow-up.
[20]	755	51	483	Low back pain, chronic fatigue syndrome	1990 American College of Rheumatology (ACR) criteria; 1994 Centers for Disease Control and Prevention criteria	Enhancing pain reconceptualization seems to be an important process to facilitate patients’ ability to cope with their condition.
[9]	428	46.5	293	Low back pain, fibromyalgia, lumbar radiculopathy awaiting lumbar surgery	1990 American College of Rheumatology (ACR) criteria	Reduction in pain intensity at 3-month follow-up, decreasing catastrophic thought processes about pain and injury.
[21]	1585	54.3	1135	Fibromyalgia, chronic fatigue syndrome, and bodily pain	NA	Effects on pain intensity and psychological distress at different time points.
[22]	460	45	325	Nonspecific chronic spinal pain, low back pain, chronic idiopathic neck pain	NA	Improvements in pain, disability, kinesiophobia, and pain catastrophizing.
[23]	1085	45.92	863	Chronic musculoskeletal pain and central sensitization	1990 American College of Rheumatology (ACR) criteria; Centers for Disease Control and Prevention for CFS (1994)	Improves pain, disability, and psychosocial factors in patients.
[24]	1664	46.5	1332	Low back pain, fibromyalgia, chronic neck pain	1990 American College of Rheumatology (ACR) criteria	Better than minimal intervention for pain intensity and disability in the short term, and for pain intensity in the intermediate term.
[25]	NA	NA	NA	Chronic musculoskeletal pain, low back pain	1990 American College of Rheumatology (ACR) criteria	A statistically significant impact on all the explored pain outcome measures.
[26]	2352	48.3	1958	Chronic neck pain, fibromyalgia, spinal pain, Achilles tendon pain	1990 American College of Rheumatology (ACR) criteria; Tampa Scale for Kinesiophobia; Pain Catastrophizing Scale; Hospital Anxiety and Depression Scale	Linear relationship between longer duration of PNE (total minutes) and reduction of anxiety symptoms, catastrophizing, and kinesiophobia.
[27]	63	47.6	41	Chronic low back pain	Roland Morris Disability Questionnaire; Patient-Specific Functional Scale; Survey of Pain Attitudes (revised); Pain Catastrophizing Scale	Long-term effects on the primary outcome measures for this patient group.
[28]	300	44	180	Chronic low back pain	Roland Morris Disability Questionnaire	Small to moderate effect on pain for these patients.
[29]	615	45.8	382	Chronic low back pain	1990 American College of Rheumatology (ACR) criteria	Moderate evidence that the addition of PNE improves disability in the short term.
[30]	1019	NA	NA	Chronic low back pain	Tampa Scale for Kinesiophobia; Pain Catastrophizing Scale	Short-term carryover effect on pain intensity and pain cognition, and a long-term carryover effect on kinesiophobia.
[31]	1078	NA	NA	Low back bain	NA	Adding PNE to treatment programs lead to more efficacious effects.
[32]	601	NA	NA	Chronic pain	Roland Morris Disability Questionnaire	PNE appears to be effective (by reducing disability) as a sole intervention for adults with chronic pain and only immediately after the intervention.
[33]	622	NA	428	Chronic nonspecific spinal pain	NA	Low evidence that PSE plus exercise therapy reduces pain, disability, kinesiophobia, and catastrophizing.
[34]	NA	NA	NA	Osteoarthritis	1990 American College of Rheumatology (ACR) criteria; 1994 Centers for Disease Control and Prevention criteria	An improvement in the groups managed with PNE, finding a small effect in favor of the interventions for variables such as kinesiophobia.
[35]	612	NA	410	Fibromyalgia	Fibromyalgia Impact Questionnaire; Pain Catastrophizing Scale; Hospital Anxiety and Depression Scale	PNE can be an effective approach for improving functional status, pain-related symptoms, anxiety, and depression.
[36]	479	NA	249	Chronic neck pain	1990 American College of Rheumatology (ACR) criteria	PNE effectively reduced pain intensity and kinesiophobia.

**Table 2 brainsci-14-00947-t002:** Characteristics of included neuroimaging studies.

Author	Participant	Age	Gender (F)	Diagnosis	Diagnostic Criteria	Neuroimaging Outcomes
[13]	1	36	1	Chronic low back pain	1990 American College of Rheumatology (ACR) criteria	Reduction in activation in primary somatosensory cortex, anterior cingulate cortex, parietal cortex and frontal cortex.
[39]	1	30	1	Low back pain	1990 American College of Rheumatology (ACR) criteria	Increased activation in the motor cortex.
[40]	120	39.9	38	Chronic spinal pain	NA	Gray matter morphologic features did not change in response to treatment.
[41]	111	36.9	69	Chronic musculoskeletal pain	Roland Morris Disability Questionnaire	Several structural and functional changes predominantly in prefrontal cortical regions.
[42]	63	42.6	45	Chronic whiplash-associated disorders	NA	Patients had decreases in GM volumes of the left and right central operculum and supramarginal after treatment.

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
