# Peer review of "Pain Neuroscience Education and Neuroimaging—A Narrative Review"

_brainsci, 2024, doi:10.3390/brainsci14090947_

Round 1

Reviewer 1 Report

Comments and Suggestions for Authors

Dear authors, thanks for raising the interest on such an interesting and actual topic. However, some clarifications are needed. First of all, many sentence are not supported by appropriate reference. In line 43-44 of the introduction the author talks about tissue damage, but in chronic pain often there is no tissue damage supporting the pain presentation, that should be clearly stated. I found that the content and aims of pain education should be clearly described in the introduction section. Further, the discussion is mostly a repetition of previous findings (should then be moved in the introduction section), and should be focussing more on explanation and important clinical application as well as possible future directions to improve the use of PNE in usual practice. In line 307 it refers to "therapy" but it was not specified before which therapy was applied in that study. Finally, the part of the discussion on brain changes is too short, and does not reflect the title of your paper.

Comments on the Quality of English Language

I suggest to authors to have a native english person checking the manuscript.

Author Response

Dear authors, thanks for raising the interest on such an interesting and actual topic. However, some clarifications are needed.

I thank the Reviewer for the interest in my work and for all the valuable feedback provided. I believe these comments helped me to clarify several points and significantly improved me manuscript.

I have addressed all the general and specific comments as described below, and revised the manuscript to improve its readability.

First of all, many sentence are not supported by appropriate reference.

Thanks for suggestion, I added references

In line 43-44 of the introduction the author talks about tissue damage, but in chronic pain often there is no tissue damage supporting the pain presentation, that should be clearly stated.

that's right, thanks for the suggestion. I changed the sentence.

I found that the content and aims of pain education should be clearly described in the introduction section.

I changed the introduction to make the explanation of PNE clearer.

Further, the discussion is mostly a repetition of previous findings (should then be moved in the introduction section), and should be focussing more on explanation and important clinical application as well as possible future directions to improve the use of PNE in usual practice.

I have modified the discussion following the reviewer's suggestion.

In line 307 it refers to "therapy" but it was not specified before which therapy was applied in that study.

Thanks for the correction, it was an inaccuracy. I corrected it.

Finally, the part of the discussion on brain changes is too short, and does not reflect the title of your paper.

I have modified the discussion following the reviewer's suggestion.

Reviewer 2 Report

Comments and Suggestions for Authors

The authors have developed a well-conducted and well-written review to to discuss what Pain Neuroscience Education is, how it is administered, and its effectiveness as a treatment for musculoskeletal pain.

However, I would like to make a few observations before recommending their work for publication.

1. Why didn't the author consult other databases such as WoS or Scopus?

2. Although your paper deals with imaging tests to assess the neural impact of PNE, I advise the author to discuss a recent landmark paper on PNE: DOI:10.3390/ijerph19106194

 3. I ask the authors to detail the selection process during the search for articles: Total number of articles found, articles selected by reading the title and abstract once duplicates are excluded, and articles finally included in the review once the complete reading of the articles has been carried out.

Comments on the Quality of English Language

No comments

Author Response

The authors have developed a well-conducted and well-written review to to discuss what Pain Neuroscience Education is, how it is administered, and its effectiveness as a treatment for musculoskeletal pain.

I thank the Reviewer for the interest in my work and for all the valuable feedback provided. I believe these comments helped me to clarify several points and significantly improved me manuscript.

I have addressed all the general and specific comments as described below, and revised the manuscript to improve its readability.

However, I would like to make a few observations before recommending their work for publication.

  1. Why didn't the author consult other databases such as WoS or Scopus?

I also searched Scopus and WoS, but it did not produce additional results. However, I added these two databases in the methods in paragraph 2.3.

  1. Although your paper deals with imaging tests to assess the neural impact of PNE, I advise the author to discuss a recent landmark paper on PNE: DOI:10.3390/ijerph19106194

Thanks for the suggestion, even though this review is not about neuroimaging, but about the effects of patient education on function. I still added it to the bibliography because it enriches the debate on the topic.

  1. I ask the authors to detail the selection process during the search for articles: Total number of articles found, articles selected by reading the title and abstract once duplicates are excluded, and articles finally included in the review once the complete reading of the articles has been carried out.

Thanks for the question. I described this part in study selection, but I probably wasn't detailed in the description. That's why I added this part:

“Originally 1065 studies were identified through the database search. Once duplicates were removed and the titles and abstracts of all remaining unique articles were analyzed, 166 full-text articles were analyzed to verify their eligibility for inclusion in the present study. Of these 166, 139 of these articles were excluded, so twenty-seven studies were finally selected for the last check, in which 5 of these were excluded. The total number of articles that met the inclusion criteria was 22.”

Reviewer 3 Report

Comments and Suggestions for Authors

This is a nice review of PNE as a treatment or adjunct treatment approach for chronic pain. Few remarks

1. Fig 1 is very difficult to read. It has to be of high resolution.

2. The whole manuscript seems to be a wall of words. I would recommend adding a pictorial description of each section, or a table of sec 3.1 and 3.2 which would make it easy to peruse. 

3. It would also benefit to split sec 3.1 into subtopics, example, based on whether PNE helped with pain outcome or other psychological metrics. Right not there is a lot of information and it's hard to take away anything the way it is organized.

4. For sec 3.2, inserting a figure showing the brain areas affected by PNE would help.

5. the author points out further fMRI studies are needed, what about electrophysiological studies using EEG etc. looks like there is scarcity of such findings. 

Author Response

This is a nice review of PNE as a treatment or adjunct treatment approach for chronic pain.

I thank the Reviewer for the interest in my work and for all the valuable feedback provided. I believe these comments helped me to clarify several points and significantly improved me manuscript.

I have addressed all the general and specific comments as described below, and revised the manuscript to improve its readability.

Few remarks

  1. Fig 1 is very difficult to read. It has to be of high resolution.

I changed the figure to one with better resolution.

  1. The whole manuscript seems to be a wall of words. I would recommend adding a pictorial description of each section, or a table of sec 3.1 and 3.2 which would make it easy to peruse. 

Thanks for suggestion, I added table 1 and table 2

  1. It would also benefit to split sec 3.1 into subtopics, example, based on whether PNE helped with pain outcome or other psychological metrics. Right not there is a lot of information and it's hard to take away anything the way it is organized.

I thank the reviewer for the suggestion, I tried to separate paragraph 3.1 into two subparagraphs, but all the articles were linked by various aspects, so it was impossible for me to create subparagraphs. I hope that the table will be enough to make my manuscript more readable.

  1. For sec 3.2, inserting a figure showing the brain areas affected by PNE would help.

Thanks for suggestion, I added fig_2 in the manuscript

  1. the author points out further fMRI studies are needed, what about electrophysiological studies using EEG etc. looks like there is scarcity of such findings. 

It’s true, I added a sentence in conclusion.

Round 2

Reviewer 2 Report

Comments and Suggestions for Authors

The authors have improved with their current version the previous version of their manuscript, so I recommend its publication. Congratulations.

Comments on the Quality of English Language

No comments

Reviewer 3 Report

Comments and Suggestions for Authors

good to accept.